# Strategic Possibility Routes of Recycled PET

**DOI:** 10.3390/polym13091475

**Published:** 2021-05-02

**Authors:** Ho-Shing Wu

**Affiliations:** 1Department of Chemical Engineering, Institut Teknologi Sumatera, Lampung Selatan, Lampung 35365, Indonesia; damayanti@tk.itera.ac.id; 2Department of Chemical Engineering and Materials Science, Yuan Ze University, 135 Yuan-Tung Road, Chung-Li, Taoyuan 32003, Taiwan

**Keywords:** polyethylene terephthalate, pyrolysis, hydrolysis, methanolysis, glycolysis, phase transfer

## Abstract

The polyethylene terephthalate (PET) application has many challenges and potential due to its sustainability. The conventional PET degradation was developed for several technologies to get higher yield products of ethylene glycol, bis(2-hydroxyethyl terephthalate) and terephthalic acid. The chemical recycling of PET is reviewed, such as pyrolysis, hydrolysis, methanolysis, glycolysis, ionic-liquid, phase-transfer catalysis and combination of glycolysis–hydrolysis, glycolysis–methanolysis and methanolysis–hydrolysis. Furthermore, the reaction kinetics and reaction conditions were investigated both theoretically and experimentally. The recycling of PET is to solve environmental problems and find another source of raw material for petrochemical products and energy.

## 1. Introduction

The most significant application of polymers in the last two decades is polyethylene terephthalate (PET), which is of excellent chemical and physical properties for many implementations, for instance, characteristic of the gas barrier, low diffusivity, excellent mechanical and thermo mechanical properties, highly inert material, clearness and fine process operation [1,2,3,4]. On the other hand, PET waste is already highlighting for humans and the environment. The global cumulative amount of plastic waste generated from 1950–2015 was approximately 6.3 billion tons, around 9% of which had been recycled, 12%was incinerated and 79% was accumulated in landfills or the natural environment [5]. Each year, the forecast PET waste about million metric tons transfers into the ocean and landfill. Currently, the recycling plastic methods are landfill, incineration and energy recovery and plastic recycling. The conventional landfill and incineration methods were concerned since the plastic component can be released into the ambient environment during processing. The landfill and incineration methods have hazardous substances released into the environment [6,7,8]. Most plastics are nondegradable, and take a long time to degrade, probably takes hundred years; however, nobody knows precisely when the plastic is degraded at the landfill. Based on the Environment Protection Agency, plastic recycling is divided into three-part, (i) be used directly, (ii) be undergone physical reprocessing, for instance, grinding, melting and reforming, and (iii) be undergone chemical processing when components are isolated and reprocessed for use in the chemical industry [9,10,11].

Several strategies can be applied to reduce the waste of PET until 2040. The approach to zero plastic pollution is divided into four critical types of interventions: reduce, substitute, recycle and dispose. Moreover, the eight actions can be implemented: (1) minimize the quantity of single used plastic (2) replaced the petroleum plastic with the other variant of materials and delivery systems (3) implementing design for recycling (4) raising the capacity of the collection (5) enlarge the capacity of sorting and mechanical recycling (6) increasing chemical conversion capacity (7) minimize post-collection environmental leakage (8) the trade of plastic become decrease slightly [12]. PET has a low modulus of synthetic fibers, its properties closeness to the other polymer, for instance, polyethylene, nylon and polyester [13,14]. The recycling process of PET can be conducted using mechanical and chemical processes. The primary purpose of recycling PET is to modify the polymer of PET into economically reusable forms. PET chemical recycling was broadly used in the chemical products such as polyester molding compound, varnishes, polymer plaster, topcoats of reinforced plastic, mortar and mineral filler, fiber, polyol for polyurethane elastomer, polyurethane with low flammability and foam [15]. The primary purpose of recycling PET to modify the polymer of PET into economically reusable forms; moreover, the critical point out of recycling PET is not only to reduce the cost of process production but also to maintaining an ecological balance is essential for sustainable to save our planet [16,17,18]. However, the cost of the chemical recycling of PET is higher than that of physical recycling of PET. For that reason, the innovation of technology to the chemical recycling of PET is needed.

This review aims to show the recent technology of chemical recycling PET to reduce PET waste in the environment. Several researchers have already developed the chemical recycling PET to replace landfill and incineration methods. Moreover, this work focused not on machinal recycling but also chemical recycling of PET such as degradation, hydrolysis process by alkaline and acid hydrolysis, methanolysis, glycolysis, microwave, ionic liquid, phase–transfer catalysis, combination glycolysis-methanolysis, glycolysis–hydrolysis, methanolysis–hydrolysis. Furthermore, the kinetic study of the chemical recycling of PET is reviewed to find more information; it can make the end-product of recycling PET more sustainable for the environment.

## 2. Physical Properties PET

PET is a popular plastic resin and a common type of polyester used commercially. The first synthesis of PET in 1942 by Whinfield and Dickson was carried out in the United Kingdom at the beginning of World War II, whereas under the Calico Printers Association. Furthermore, the fiber melt–spam from the new polyester was produced by Imperial Chemical Industries-United Kingdom, under the commercial name TERYLENE. Moreover, the DACRON is manufactured from DuPont, United States, in 1953 [19]. Since that time, PET took place as the most massive production of synthetic fiber on our planet. At the end of the 1970 s, the stretch molding process was developed as a bulk chemical to produce PET into long-lasting crystal-clear beverage bottles. This application took second place as the most application used in fiber production [20].

The product from the PET recycling process has increased. Europe is one of the top leaders for recycling PET. The average rate of recovering plastic containers in the European Union (EU) was 26%. Still, some nations, for instance, Sweden and Belgium, reached 40% by 2013. According to the US, plastic packaging was recycled about 12% in the US Environmental Protection Agency 2010. Japan is one of the most efficient collection systems for PET bottles globally, with 72.1% had been recycled [21].

The worldwide production of PET in 2017 was up to 30.3 million tons. The largest PET producer is dominated by China, which a total output was 30.8% around the world. Furthermore, the production for the Asian region except for Mainland China up to 21% from the production share. The leading country in the North American region for PET production is the United States, with a total share of 16.9%. Currently, the MG Chemicals of Corpus Christi, Texas, US, build a new plant with the capacity of producing 1.1 million tons per year. The European membership calculated for 14.7% of the total production capacity of PET, followed by the Middle East (10.2%), South America (4.1%) and Africa (2.3%). The commercial PET has a melting temperature (Tm) of between 255 and 265 °C, and for more crystalline PET is 265 °C. The glass transition temperature (Tg) of virgin PET varies between 67 and 140 °C [22], which is affected by the procedure of measurement and the polymer’s state. The chip sample of PET gives a value of ~78 °C measured by differential scanning calorimetry (DSC). Still, a highly oriented and crystalline drawn fiber measured by the dynamic loss method will provide benefits with as high as 120 °C. Furthermore, the specific gravity of the amorphous PET is 1.33. In contrast, crystalline drawn fiber has a value of 1.39 [19]. The crystallization of PET had the slowest crystallization rate with the range temperature crystallization at 170–190 °C [22,23]. The range of heat fusion is ~140 J/g (33.5 cal/g) [19].

## 3. The Common Recycling of PET

The recycling processes are one way to reduce PET waste; moreover, most PET wastes came from bottles and food containers. Furthermore, the virgin PET remains stable, fresh and cheaper than recycling PET [24]. The PET can be recycled to initial raw materials such as terephthalic acid (TPA), dimethyl terephthalate (DMT) and methyl ethylene glycol (MEG), where the polymer structure is eradicated or can be recycled back into its process intermediates like bis(2-hydroxyethyl) terephthalate (BHET). The rear part shade light on various techniques to produce TPA from PET [25]. On the other hand, the critical factor affecting post-consumed PET flakes’ suitability for recycling is the level and nature of contaminants present in flakes [24]. Contamination is one of the problems to recycling post- consumed PET. Moreover, it caused deterioration of physical and chemical properties during reprocessing [26]. Dimitrov et al. already studied the effect of contaminant to post consumed PET with thermal degradation method. The major product PET (contaminated flakes) is carbon dioxide/acetaldehyde, and 4-(vinyloxycarbonyl) benzoic acid, with the total percentage of products, are 43.9% and 3.66%, respectively. The low molecular product (CO_2_, acetaldehyde) is the most by-product; it is caused by accelerated pyrolytic degradation due to the catalytic effect of remained contaminants (like D-limonene, chlorobenzene, benzophenone) [27]. There are several contaminants that can be affected on recycle PET, such as: (i)Water Contaminant

Water is one of the contaminants that lead to the hydrolytic chain cleavage of PET. For instance, the polymer should be rigorously dried before melt reprocessing. The temperature drying to recover PET flakes is 160–180 °C. Furthermore, the moisture content is devolatilized before PET becomes molten rapidly reacts. Then, the small amount of moisture can decrease the polymer’s viscosity to such a level that acceptable bottles cannot be blown [28]. The initial rate PET contributes to the hydrolysis process. It caused the waste of water in the sample, while the later slow rate is attributed to the thermooxidative chain depolymerization by thermal energy [29].
(ii)Coloring Contaminant

A small quantity of pigments is added. The copper phthalocyanine blue is applied as a colorant for food packaging and containers. Benzotriazole UV stabilizers are added to rPET to keep safe for some kinds of food from the sunlight. Tinuvin 326 is applied as addictive to protect edible oil against photooxidation [30]. The presence of contaminants created some issues, such as cleavage of chains, raising the monomer of carboxylic end groups, reducing molecular weight and decreasing intrinsic viscosity, leading to a decrease in mechanical properties of the material [31].
(iii)Acetaldehyde

Acetaldehyde is present in PET and post-consumed PET. It is a side product of depolymerization of PET reactions [24]. The reaction occurs because recombination of vinyl ester and hydroxyl end groups of PET. Simultaneously, this kind of reaction can be produced vinyl alcohol such as acetaldehyde by the automerization process. The acetaldehyde will be transferred into a final product, which is an essential issue in food application.

On the other hand, acetaldehyde relatively volatile. Therefore, it can be minimized by drying or vacuum process. The kind of stabilizer addictive can be mixed to minimize the acetaldehyde contain, such as 4-aminobenzoic acid, diphenylamine and 4,5-dihydroxybenzoic acid [32].
(iv)Heavy Metal Contamination

The five heavy metals for recycling PET are led nickel, cadmium, antimony and chromium. These kinds of five metals influence serious health problems with extreme exposure. The sources of heavy metal contamination can be labels, adhesives, inks and debris during transport and sorting [33].

### 3.1. Conventional Recycling PET

The typical recycling of PET is by water-based washing of the post-consumer PET to remove and wash the top side of contamination, dirt, labels and glue. Figure 1 shows the scheme of the conventional recycling process of PET. The first step of recycling PET is the post-consumer PET bottles resized, and it became flakes. The common addictive as a cleaning agent of PET used are caustic soda with the percentage of usage about 2–3% and detergents [34]. The second step of recycling PET is the mechanical reprocessing process. Generally, the PET waste retreated into granules by conventional extrusion after removing it from the contaminants. Collection/segregation, cleaning and drying, chipping/sizing, coloring/agglomeration and palletization/extrusion are the subsequent processes before manufacturing the end product [35,36]. The tertiary recycling process is the depolymerization process of PET to become monomers or other valuable low molecular weight monomers. Re-polymerized can achieve the monomer from the initial polymer. The next step is the crucial stage to become most profitable and beneficial from a sustainability perspective. Furthermore, it could reduce the demand for energy and raw material cost [35].

### 3.2. PET Super-Clean Recycling Processes Based on Pellets

The first stage of the super clean process focuses on removing all the contaminants attached to the surface of PET. Furthermore, the PET re-extruded becomes pellets size. The solid-state polycondensation (SSP) technology is used to further deeply clean PET. The SSP can be operated with a batch or continuous process. The operation parameters of this process are residence time, temperature, vacuum, inert gas stream. Commonly, the range of residence time for solid-state reactions is 6–20 h, depending on temperature reaction and the desired PET material’s viscosity. The temperature range for this situation is 180–220 °C. Figure 2 shows the scheme of PET super-clean recycling processes based on pellets. In the fact that the process of recycling of PET for the super-clean method is very close to virgin processes whereas its same apparatus. Due to the re-extrusion process, there are some advantages to recycling PET. For instance, the pellet size of virgin PET is uniform, and the contamination of post-consumer PET is homogeneously distributed [34].

## 4. The Mechanical Recycling of PET

Some research already implemented PET recycling by mechanical or chemical methods into their needs. The various waste of PET through mechanical recycling can be found in packaging, films, containers, sheets, fiber for insulation and floor covering [37]. The current technology process for the first and second stages are dry processes without using washing water. The rPET process consists of classifying, sorting, washing and drying, size reduction, melt filtration, reforming and compacting steps due to the diversity of source plastic materials [38].

The waste of PET is passed to a pretreated process, which is the sorting process. It is carried out according to the best conditions, such as screening waste of plastic based on the size of particles, fines, heavy and light materials and separation of waste plastic according to the characteristic of shape, e.g., 2D and 3D particles. Terzini et al. studied the apparatus for municipal processing waste. Figure 3 shows the step of municipal processing waste. The conveyor carries out the waste material into the shredder with organic and inorganic materials. The next step is divided into two streams; the first stream will be used for large particle streams, for instance, eliminated light, ferrous, aluminum, plastic and combine particle streams. The other stream is to remove a small particle stream, such as to remove ferrous, aluminum and combinations of particle streams. The rotary screening is applied in this stage [39].

The next stage is crushing. The bottle would be classified based on its properties. On the other hand, this kind of method would decrease the particle size. Furthermore, it makes a vast number of fine and non-sortable particles. The dust particles can affect the product quality [38]. The best illustration for slicing machine by the Valley beater and the disc refiner is that the PET waste must pass within a plurality of knife edges in intermittent sliding contact. This process needs up to 10 min. Furthermore, the average residence mixing time is 1 min or less for a disc refiner’s continuous operation [40].

Counter comb shredders are commonly used for crushing raw materials, especially for bedrocks or waste and recycling sectors. This kind of machine had a minimum rotatably driven comminution roller. The comminution apparatus such as teeth, cutting edges and/or movable hammers are provided on the comminution roller’s cylindrical shell, i.e., on the roller body. The comminution process started with the counter apparatus of the comb. Typically, the comb is designed like a comb beam for holding the device, and it commonly extends over at least the entire width of the comminution roller.

Moreover, the comb can be configured as a single or multi-piece unit [41]. Furthermore, the rotary drum screen is commonly used in industrial plastic waste. The advantage of the rotary drum screen is straightforward to operate the waste plastic with a large amount of waste plastic. Furthermore, it can be to reduce the consumption of energy. The range of drum diameters 2 m to 4 m with the suitable diameter for domestic waste is approximately 3 m. The plastic waste is carried to the upper part of the rotary sieve as it turns by carrier plates that are axially attached [42].

The single plastic solid waste had a different way to recycle by a mechanical process. The re-bonding process used a recycled foam flake that comes from flexible slab stock foam production waste. The slab stock foam is stored in the silo. After that, the stock foam flows into a mixer consisting of a fixed drum with a stirrer or rotating blades. The foam flakes are sprayed with an adhesive mixture. This process’s primary benefit is to get a clean product with recent properties, for instance, higher density and lower hardness [43] at the recycling process of PET with a mechanical process. It is applied with a different stage of the process such as assorted, ground, washed and extruded. The degradation of polymer is used in the reprocessing of polymers, and it required several degrees of polymer degradation [44]. The technology of recycling PET by mechanically processing with thermal such as:
a.Extrusion molding: The small particle of plastic solid waste is melted and extruded through by single or twin screws to get the molded product—applying extrusion moldings such as pipes, sheets, film and wire covering.b.Injection molding: The melted polymer is injected into a mold to solidify form to obtain the product desired. The implementation of this process, for instance, washbowls, buckets, bumpers and pallets.c.Blow molding: Air is applied in the blow molding. The extrusion or injection molded is clamped in a mold with air to make bottles or containers.d.Vacuum molding: Mild heat layer is sandwiched in the mold, and there is a space between the layer and mold sealed. The vacuum moldings are used for several products, e.g., cups and trays.e.Inflation molding: The melted polymer is inflated into a cylinder to form a film by extrusion molding—the implementation of inflation molding such as shopping bags [44].

Mechanical recycling is more practiced but results in the degradation of the plastic properties, which high-quality end-products is achieved using the chemical recycling method. Mechanically recycling waste PET into new bottles requires pristine, transparent raw materials. Moreover, PET degrades every time it is reprocessed; after about six cycles, it is no good [45].

### Contamination Levels and Maximum Consumer Exposure from Food Packages Made from rPET

Acetaldehyde and limonene are typical compounds derived from PET and prior PET bottle contents (flavoring components). Their materials analyzed by gas chromatography revealed average and maximum levels in rPET of 18.6 and 86.0 mg/kg for acetaldehyde and 2.9 and 20 mg/kg for limonene, respectively [46]. Maximum levels in rPET of natural contaminants such as misuse chemicals like solvents ranged from 1.4 to 2.7 mg/kg and statistically were shown to result from 0.03% to 0.04% of recollected PET bottles. Under consideration of the cleaning efficiency of super-clean processes as well as migration from the bottle wall into food, the consumer will be exposed at maximum to levels <50 ng total misuse chemicals per day. Therefore, rPET materials and articles produced by modern super-clean technologies can be considered safe indirect food applications in the same way as virgin food-grade PET.

## 5. The Chemical Recycling of PET

The conventional plastic stable carbon-based backbone makes the plastic resistant to depolymerization under different types of environmental conditions. Since plastic has good properties of materials, the production of plastic is increasing in the current decade. Furthermore, the plastic product’s significant application is short-term living, disposable packaging materials becoming waste after a single-use. Therefore, an enormous quantity of plastic waste is accumulating in the environment. Moreover, it already a severe problem of waste plastic because of poor biodegradability with detrimental effects to terrestrial and marine ecosystems and eventually on humans. On the other hand, plastic has many negative impacts, and the researchers try to develop some method/technology to recycling the PET [47,48,49].

Chemical recycling involves depolymerization, purification and then repolymerization. The best conceptual recycling of post-consumer waste is to modify the PET into their monomer as chemical raw material. The remarkable chemical properties of a polymer depend on designated use. PET is a partial–crystalline structure. The properties of the crystalline structure are high strength and tensile. There are so many technologies to recycle PET to obtain the recycled PET (rPET), for instance, packaging, textile and electronic compounds. Dias et al. studied the effect of thermal decomposition of rPET. The activation energy of rPET was 193 CV/Kj mol^−1^. The film absorption spectrum of rPET shows the features of 1722 cm^−1^ related to carbonyl esters (C=O), within 1024 and 1259 cm^−1^ regarding with ester (CO) bond, at 731 cm^−1^, p-substitution of the aromatic ring conjugated with the carbonyl at 2973 cm^−1^, stretching of the CH bond [50]. The alcoholysis of PET was heated with superheated methanol vapors. The principal product distributions are the combination of DMT, phthalate derivatives and alcohols. The complicated product distribution makes the separation process of the alcoholysis process more expensive. Furthermore, the catalyst should be deactivated to prevent transesterification of DMT with EG into diethylene glycol terephthalate and PET [51].

Some of the organic bases, e.g., 1,5,7-triazabicyclo[4.4.0]dec-5-ene(TBD), 1,8-diazabicyclo[5.4.0]undec-7-ene (DBU) and 1,5-diazabicyclo[4.3.0]non-5-ene (DBN), can be applied to depolymerization of PET to small molecules of monomer. The glycolysis process’s operation conditions are at high pressure and temperatures up to 1.5–2 MPa and 250 °C, respectively. TBD was used as excess to convert PET to BHET, with a yield of up to 78% [52]. Furthermore, the PET chemical recycling based on environmentally friendly was developed by Pulido et al. In contrast, the separation process could occur from traditional methods such as distillation, crystallization. The range of ideal pore size rPET membrane was 35–100 nm with non-solvent-induced phase separation in ethanol or methanol. In addition, the resistances of membranes could be elevated the acid and oxidative media, for example, dimethylformamide at 100 °C [53]. The other advantage of recycling waste PET is used to active electrochemical material for energy storage. Some chemical processes are applied, such as the dissolution of rPET, fiberization through electrospinning and carbonization by a furnace [54].

Chemical recycling includes various methods such as glycolysis, methanolysis, hydrolysis, ammonolysis and aminolysis, which are usually carried out at high temperatures and in the presence of catalysts. Figure 4 shows that the possible reaction to recycling terephthalate [55]. The technologies of chemical recycling can be categories into two parts which are (1) the technologies that the polymers in the PET are dissolved as a long chain of the polymer and (2) the other methods are crack the chemical bonds within atoms in the polymer chains. In condensation polymers, for instance, polyesters and polyamides (PAs), the standard approach is to crack the ester or amide long chain. In contrast, in polyolefins (polyethylene and polypropylene), one has to break the relatively stable carbon chain [56].

The main objective of the chemical recycling of PET is to degrade the PET completely into several monomers, e.g., TPA, DMT, BHET and EG. Paszun and Spychaj [51] already present the advantages and disadvantages of the chemical recycling methods of PET. The depolymerization of PET is taking place to reverse the chemical reaction of the PET formation route. Furthermore, the degradation of PET can be degraded into its monomer or other chemical substances [44]. Moreover, the implementation of chemical recycling was derived with four groups used as (1) refinery raw materials, (2) fuel production, (3) petrochemical and (4) chemical upcycling [57]. PET is very vulnerable to chemical degradation. The eight possible reactions to recycling PET are shown in Figure 4. Table 1 lists the top industrial rPET production and their location for chemical recycling.

### 5.1. Recycle PET Using Degradation Process/Pyrolysis

The pyrolysis process is the thermochemical recycling of PET, in which high temperature and pressure are applied in the pyrolysis process. Furthermore, the pyrolysis process is used without oxygen. The technology is also called dry distillation. The concept of the pyrolysis process is the solid material is first turned into a pyrolysis reactor. Vapor from the top product of the pyrolysis reactor is condensed into a condenser. Then, the liquid is stored in the accumulator tank. The temperature range of recycling plastic is typically 370–420 °C. The liquid recycling product is a polymer fragment that changes into smaller hydrocarbon such as TPA and EG [44,70,71]. Singh et al. studied the waste HDPE, PP, PS and PET co-pyrolysis under high heating conditions with a temperature range of 420–550 °C. The distribution product by an individual polymer presented alkane and alkene groups with small oxygen-containing groups except for PET, including a large quantity of CO, CO_2_ and oxygen-containing groups [72].

The concept of the pyrolysis process is shown in Figure 5. Commonly, the pyrolysis process used a tube or autoclave reactor. The beginning of the pyrolysis process started with small particles of PET and catalyst through to reactor with the range temperature is typically 370–500 °C, the condensate of liquid product through the cold trap and the gas product goes to gas washing bottle and the gas product was collected in the same time. Cepeliogullar had been studied the depolymerization waste of PET with the temperature reaction up to 500 °C and the small liquid yield quantity could be obtained 23.1% with the major products for instances, 1-propanone, benzoic acid, biphenyl, diphenylmethane, 4-ethylbenzoic acid, 4-vinylbenzoic acid, fluorene, benzophenone, 4-acetylbenzoic acid, anthracene, biphenyl-4-carboxylic acid, 1-butanone, m-terphenyl with the activation energy; at the range temperature of 373–443 and 448–503 °C were 347.4 and 172.6 kJ/mol, respectively, by fixed bed reactor [73]. Additionally, Du et al. studied the slow pyrolysis of PET carpet face fiber and backing material with CaO as a catalyst. The yield of liquid products for face fiber and backing materials without catalyst was 23.5% and 25.9%, respectively. On the other hand, the yield of face fiber and backing material would be increased slightly with added CaO as a catalyst were 37.03% ± 6.42% and 33.43% ± 2.42%, respectively [74].

Ozsin and Putun studied the co-pyrolysis of lignocellulosic biomass with PET. The walnut shell (WS) and peach stonen (PST) were carried out with a fixed bed reactor with the temperature reaction 400–700 °C. The experimental percentage yield of WS/PET and PST/PET were 28.90 and 23.45 wt%, respectively. On the other hand, the WS/PET and PST/PET yield were 22.30 and 21.05 wt%, respectively. The major products were benzene carboxylic acid and vinyl benzoate [75]. The co-pyrolysis process of paper biomass with polymers mixture, e.g., such as high-density polyethylene, polypropylene and polyethylene terephthalate under catalyst was studied by Chattopadhyay et al. that the total percentage hydrogen production through biomass and polymers mixture with the ratio 5:1 under 40%Co/30%CeO_2_/30%Al_2_O_3_ as catalyst up to 47 Vol%. Furthermore, the catalytic performance increased slightly with the rise of cobalt loading, and the percentage of catalyst are 40%Co/30%CeO_2_/30%Al_2_O_3_ [76].

The application of the post-consumer plastic waste process in the industry is the BP polymer cracking process. The plant was established in Scotland with a capacity of 25,000 ton/year with the composition of raw materials polyolefins, polystyrene (PS), PET, polyvinyl chloride (PVC): 80 wt %, 15 wt %, 3 wt% and 2 wt%, respectively. This process was applied for size reduction for the feed. The temperature of the fluidized reactor was up to 500 °C in the absence of air. The polymer chain will already be vaporized and left the bed with fluidizing gas [77,78,79].

Park et al. reported the effect of carbon-supported Pd Catalyst in PET pyrolysis. The products from catalytic pyrolysis are polycyclic hydrocarbon and benzene derivatives. The thermochemical reaction was occurred by a free radical mechanism and ring-opening reaction. On the other hand, the more Pd catalyst loading as a catalyst would be affected less by amine product [80]. Furthermore, the pyrolysis process’s disadvantage is the harmful material, such as polycyclic compounds and biphenyls, that caused a severe environmental problem. The polycyclic aromatic compounds (PAHs) can modify into specific oxygenated–PAHs, which the secondary chain of organic aerosol reacts with NO_X_ and O_3_ [81,82,83]. Jia et al. reported the effect of fast catalytic pyrolysis of PET with zeolite and nickel chloride with a different temperature reaction (450–600 °C) under an N_2_ atmosphere. The yield of waxy produced from 59.5–67.7 wt% to below 10–23 wt% with various types of pyrolytic temperatures without a catalyst. Furthermore, the ZSM-S is used as a catalyst. The waxy product shows that the carbonyl and aliphatic groups, which are C-O bonds, will be declined up to 42% and 20%, respectively [84].

### 5.2. PET Recycling by Hydrolysis

There are so many methods to recycle PET by hydrolysis process, such as acid, alkaline and neutral hydrolysis [85]. The hydrolysis process’s significant disadvantages are high temperature with a temperature range of 200–250 °C and pressure up to 1.4–2 MPa. Meanwhile, the reaction time should be enough to complete of degradation of the PET process. The hydrolysis process is commonly not widely applied to food-grade rPET production due to the high cost of purification of recycled TPA (rTPA) [86]. Adding catalysts could increase the reaction rate to increase the rTPA production. The catalysts are as MCl(M: Na, Ca), MSO_4_(M: Mn, H_2_), and (CH_3_COO)_2_M (M: Zn, Cu, Co, Cd, Na, K), MOH, MHCO_3,_ and M_2_CO_3_ (M: Na, K) and HCl.

#### 5.2.1. PET Recycling by Alkaline Hydrolysis

Alkaline hydrolysis is one of the methods to recycle PET in route 5 in Figure 4. PET was depolymerized to produce EG and terephthalic salts instead of TPA at mild temperatures and pressures. Terephthalic salts can then be hydrolyzed easily to form TPA using HCL or H_2_SO_4_. Commonly, the PET recycling by alkaline hydrolysis is carried out by an aqueous alkaline solution of NaOH or KOH with a concentration range up to 4–20 wt% [87]. This kind of process is carried out slowly. Therefore, the dissociation constant K > 10^−5^ of amine can be used to accelerate the process. The time reaction needs 3–5 h with the temperature reaction up to 210–250 °C and under the pressure of 1.4–2 MPa [88].

The depolymerization of PET was studied by hydrolyzed at pH 3.0–10.5 within 4 weeks at 80 °C. The high yield of TPA and EG was found with carbonate buffer (pH 10.5) [89]. Pitat et al. presented that the PET recycling by alkaline hydrolysis was carried out with 18 wt% sodium hydroxide solution. The aqueous solutions of alkaline hydroxides are achieved as a product. The possible way to achieve complete precipitation is to keep sodium hydroxide concentration in the level 18 wt% of sodium hydroxide. After separation, it will be dissolved with water to get a saturated solution. This kind of process is the acidification process.

The alkaline hydrolysis of a mixture of PET waste and methyl benzoate produced a byproduct of the oxidation of p-xylene to TPA (route 7 in Figure 4) [55,90]. In the first stage, PET was treated with methyl benzoate at 190–200 °C. The process allows TPA and benzoic acid recovery with yields of 87–95% and 84–89%, respectively.

Furthermore, the TPA is precipitated from the solution, filtered, rinsed and dried. The other chemical component, except TPA, EG, is formed during the reaction [91]. The impact of NaOH concentration on the product distribution, such as TPA, EG, oxalic acid and CO_2_ yields, was studied by Yoshioka et al. The oxalic acid is formed by the based catalyzed EG oxygen oxidation with the first-order function of OH concentration [92].

#### 5.2.2. PET Recycling by Acid Hydrolysis

Acid hydrolysis is performed most frequently using concentrated sulfuric acid, although other mineral acids, such as nitric or phosphoric acid, have also been employed. The hydrolysis of PET by acid catalysis includes hydrochloric acid and sulfuric acid in route 4 in Figure 4. To avoid high pressure and temperature in the reaction vessel, a concentrated sulfuric acid (>14.5 M) was proposed by Pusztaszeri et al. [93]. The degradation of PET at atmospheric pressure in 3–9 M sulfuric acid below 150–190 °C for 12 h to clarify the mechanism for a feedstock recycling process. TPA and EG were produced by the acid-catalyzed heterogeneous hydrolysis of PET in sulfuric acid. The TPA yield agreed with the degree of PET hydrolysis, but the EG yield decreased with increasing sulfuric acid concentration because of the carbonization of EG [94].

Karayannidis et al. reported that the degradation process of PET using acid hydrolysis was conducted in a 2-L stainless-steel autoclave reaction vessel for temperatures above 100 °C (Pressure Reactor-Parr No. 4522). The effect of temperature had a great result in improving the yield of TPA. Interestingly, at 200 °C, the TPA yield was obtained up 98% with a reaction time of only 1 h [95]. Still, the corrosion of the equipment and the separation of ethylene glycol from the waste acid were critical problems in the process.

### 5.3. PET Recycling by Methanolysis

Route 2 in Figure 4 shows that the possible reaction of PET degradation in methanol. The degradation process of PET by methanolysis is carried out by high temperature and high-pressure conditions with the major products of DMT and EG [86,96]. The main benefit of the methanolysis process is the installation, and it can be located in the polymer production line. Furthermore, methanol and EG can be rapidly recycled. This process’s disadvantages are extremely expensive due to the separation process and refining of the mixture composition from reaction products such as glycols, alcohols and terephthalate derivatives [86].

The depolymerization of PET in supercritical methanol had been studied by Kim et al., The raw material obtained from the waste of beverage bottles. In addition, the materials from unknown chemical pretreatment. The incising temperature of the depolymerization process will be affected by the conversion of PET and DMT yield. The experiments were carried out with a temperature range of 280–310 °C. The yield of DMT, EG and the side product are 91.3, 93.9 and 94.0%, respectively, with the reaction time of 50, 60 and 70 min, respectively [97]. Methanolysis is commonly used to degrade PET using methanol at high temperatures under high pressures. However, Loop industrials claim that their depolymerization process achieves methanolysis using low heat and no pressure [67]. Pham and Cho [98] developed a low-energy catalytic route for methanolysis to convert PET resin to DMT. K_2_CO_3_ was used as a catalyst to develop a new decomposition pathway towards DMT at ambient temperature.

### 5.4. PET Recycling by Glycolysis

Glycolysis is the most popular method to depolymerize waste of PET in route 3 in Figure 4. This process already develops on a large scale. The outcome from deep glycolysis by EG is primarily BHET, by the same way to DMT, is a substrate for PET synthesis [88]. The monomer of BHET is used mainly in chemical production, such as unsaturated polyesters, rigid otherwise flexible polyurethanes and other intermediate chemical products or fine chemical raw materials [99]. The process was conducted with a temperature range of 180–250 °C with a reaction time of 0.5–8 h for the glycolysis process. In addition, the catalyst was added to the reaction, such as zinc acetate [88]. Table 2 listed the effect of kinds of catalysts on the PET glycolysis process with BHET as a particular product. Goje and Mishra indicated their optimization of the glycolysis process. The optimal size of raw material was 127.5 mm. The degradation of PET was increased with the rise temperature and reaction time. The glycolysis process was conducted in xylene by different glycols such as EG and PG with the various molar ratios of PET/glycol up to 1:0.5 – 1:3 with the temperature reaction at 170 – 245 °C and the yield of BHET produced up to 80% [100].

This PET glycolysis produces impurity components such as DEG, DEG ester and oligomers. Except for deionization and decoloring, Inada and Sado reported a purification method for obtaining BHET of high quality efficiently by subjecting an EG solution of crude BHET containing 2-hydroxyethyl(2-(2-hydroxyethoxy)ethyl)terephthalate and an oligomer as impurities to crystallization and/or membrane and molecular distillation under specific conditions (route 1, process of PET Refine Technology Co., Ltd., Kawasaki, Japan) [105]. Figure 6 shows the three possible reactions of PET recycling by glycolysis process. Raheem et al. reported a process simulation of BHET (route 3) and its recovery using two-stage evaporation systems using ASPEN PLUS software [106], which operating conditions of the reaction were PET particle size of 127.5 μm, EG: PET (*w*/*w*) ratio of 5:1, the temperature of 469 K, the pressure of 1 atm and residence time of 3 h; and the reaction results were 100% depolymerization of PET, 85.24% yield of BHET and 14.76% oligomer [107]. Goh et al. reported the optimum conditions of BHET crystallization (needle-shaped crystal) in route 2 in Figure 6 were 3 h, 2 °C and 5:1 mass ratio of distilled water used to give the highest yield and purity of the crystallization process [108,109]. The addition of hot water at 65 to 75 °C dissolves and separates BHET from the residue’s oligomer. The hot aqueous solution was cooled to 5 to 15 °C to extract the BHET. At the end of the crystallization process, hot water at 70 to 80 °C was re-added to the BHET obtained, followed by a cooling process to 5 to 15 °C for further crystallizing the BHET [110]. The solubility of BHET is around 22 g/L-water at 40 °C [110]. The ASPEN tech developed a simulation example in 2020 of the reaction conditions: PET of 1000 kg/h, EG of 5000 kg/h, ZnAc of 10 kg/h and reaction temperature of 190 °C in a batch reactor; and after 3 h, the output results of PET of 82 kg/h (conversion 91.8%) and BHET of 1222 kg/h using ASPEN PLUS software but this example didn’t discuss the effect of PET particle size [106].

### 5.5. PET Recycling by Aminolysis and Ammonolysis

Chemical depolymerization of PET bottles via ammonolysis and aminolysis in routes 6 and 8 in Figure 4 were presented by Gupta and Bhandari [111]. The aminolysis of PET yields diamides of TPA, which is known as bis (2-hydroxy ethylene) terephthalamide (BHETA). Investigation of conventional analytical methods for determining the conversion of PET waste degradation via aminolysis process was studied by Ghorbantabar et al. [112], which present the purification procedure of BHETA. A series of terephthalamides were successfully produced by rapid catalyst-free microwave-assisted aminolysis of polyethylene terephthalate (PET) [113]. The reaction time varied from 10 to 60 min. The end-product obtained after aminolysis with allylamine was further reacted with a thiol through a radical thiol-ene reaction, producing good quality films with glass transition temperature (Tg) above room temperature. Aminolytic upcycling of PET wastes using a thermally stable organocatalyst was studied by Demarteau et al., which propose an aminolytic method to depolymerize PET using various amino-alcohols by organocatalysis and their subsequent use for the production of poly(ester-amide)s [114].

## 6. The Recent Development of PET Chemical Recycling

### 6.1. PET Recycling Using Microwave Irradiation

The PET recycling by microwave irradiation takes more attention from many researchers. The recycling process by microwave method allows relatives to take a short reaction time with much higher energy to heat consumption than the conventional heating process. Nevertheless, the benefit of microwave irradiation of PET by glycolysis process is still cost-prohibitive due to the energy consumptions. The waste PET and catalysts are directly put together into the microwave reactor. Utilizing a mixed catalytic system in a microwave absorber, the glycolytic depolymerization process of PET through microwave irradiation would optimize energy efficiency [115].

The advantages of the chemical recycling of PET by microwave irradiation are to supple the energy of chemical reactions, increase the reaction rate, decrease the reaction time and increase the product’s yield. It can reach even though under milder reaction conditions. Milan et al. studied the degradation of PET by microwave with the solvolysis process. The first step of the PET process was mixed up with a microwave absorbing activator under atmospheric pressure and then the PET was melted. It continued with solvolysis processes such as acidic or basic hydrolysis, alcoholysis or glycolysis in the presence of a catalyst under continuing microwave radiation with atmospheric pressure. The product were TPA, salts or esters and EG [116].

The effect of 1-buthyl-3-methylimidazolium bromide liquid with PET glycolysis process under microwave was studied by Alnaqbi et al. The temperature and reaction time of glycolysis were 170–175 °C and 1.75–2 h, respectively. The major liquid product was BHET with the conversion and yield up to 100% and 64 wt% [117]. The combination of hydrolysis and alcoholysis with microwave irradiation can be achieved by an acidic catalyst, such as montmorillonites K10 and KSF, ion exchangers, zeolites, phosphoric acid supported on alumina or silica, copper(II), iron(III), Zinc(II), aluminum (III), antimony(III), bismuth(III) chlorides or acetates, respectively. Or it can be performed by homogeneous catalysts, such as p-toluene sulphonic, formic, acetic, benzoic, terephthalic or sulfuric acid, respectively. Liu et al explored related to catalytic hydrolysis under microwave assisted with *Brønsted* acidic ionic liquids for instances, 1-hexyl-3-methylimidazolium hydrogen sulfate ([hexanemim][HSO4]), 1-Ethyl-3-methylimidazolium hydrogen sulfate([Emim][HSO4]), N-methylimidazolium hydrogensulfate([Hmim][HSO4]), 1-butyl-3-methylimidazolium hydrogen sulfate([Bmim][HSO4]) [116,118]. The hydrolysis of PET under high pressure up to 2.8 – 3.0 Mpa investigated by Ikenaga et al with the range temperature reaction 235 – 237 °C and added 1.0 %wt as a catalyst to the major product was TPA [119].

Gr3n company technology offers a new and revolutionary approach to the chemical treatment in water systems using a microwave-assisted technology able to treat rPET in a closed-loop cycle process. The core of the technology is DEMETO (depolymerization by microwave technology), a patented technology, able to depolymerize continuously, a wide range of PET manufacturers (e.g., color bottles, food containers, polyester textile), which reduces the reaction time from 180 to 10 min [59].

### 6.2. PET Recycling Using Ionic Liquid

The ionic liquids have been developed significantly as a catalyst for PET depolymerization by the glycolysis process since the first invention in 2009. The first publication of ionic liquids was published by Zhang et al. that the halometallate catalyst based on ionic liquids for depolymerization of PET in the presence of EG. Furthermore, acidic ionic liquids’ application as dual-purpose catalysts simultaneously performs as a Lewis acid and nucleophile. It shows higher activity and selectivity than those used simple metal salts or solely organic ionic liquids [120,121,122]. Furthermore, the PET recycling by ionic liquid has several advantages: the extensive range broad of compound selection combined with anion and cation, non–volatility, thermal, electrochemical and low flammability [123,124,125,126].

The degradation of PET in supercritical ethanol with ionic liquid 1-butyl-3-methylimidazolium tetrafluoroborate ([Bmim][BF_4_]) was studied by Nenes et al. It indicated an up-and-coming method to degradation of PET in a sustained way [121,122,127]. Furthermore, Al-Sabagh et al. studied that the effect of ionic liquid-coordinated ferrous acetate complex immobilized on bentonite as a novel separable catalyst for PET glycolysis under mild conditions. The PET conversion up to 100% and the yield of BHET up to 44% with the reaction temperature at 190 °C [128].

### 6.3. PET Recycling Using Phase–Transfer Catalysis

One of the essential methods in organic synthesis is phase-transfer catalysis. In the manufacture of fine chemical, it will be easier that can be used two immiscible phases, for instance, liquid-liquid and solid-liquid, in the presence of a chain transfer agent, which can be a quaternary ammonium salt or crown ether [129,130,131,132]. The effect of quaternary ammonium salt as a phase-transfer catalyst for the microwave depolymerization of PET waste bottles had been studied by Khalaf et al. The PET was depolymerized to TPA and EG by hydrolysis process using 10% NaOH and ammonium salt as phase-transfer catalyst under microwave irradiation, which PET conversion was up to 99% [133]. López-Fonseca et al. presented that tributylhexadecylphosphonium bromide was found to be the most effective catalyst to conduct the PET alkaline hydrolysis (>90% conversion) at 80 °C and 1.5 h [134]. Kosmidis et al. presented the kinetics of PET alkaline hydrolysis by phase-transfer catalysis [135].

### 6.4. PET Recycling Using Nautral or Biomass-Based Catalyst

Stanica-Ezeanu and Matei obtained a high yield of rTPA from PET depolymerization by neutral hydrolysis in marine water [85]. Lalhmangalihzuala et al. studied the glycolysis of PET at 190 °C using biomass-waste orange peel ash as a catalyst that obtained the 79% BHET after 1.5 h [136]. Rorrer et al. investigated the combination between rPET and biomass became long life time composite materials with the properties over petroleum materials based. The rPET would be reacted with the monomer of diols from biomass. Then, the deconstruction of PET took place in this process, and the process continued with bio-derivable olefinic acid. The outcome from this process was unsaturated polyester (UPE). Furthermore, it continued with the diluent process of reactive bio-derivable, and the results were fiberglass reinforced plastic, and it could save energy up to 57% [137]

### 6.5. PET Recycling Using an Enzymatic Catalyst

Enzymatic hydrolysis offers an interesting biotechnological route to PET degradation under mild conditions but with severe limitations such as the requirement for amorphous or low-crystallinity PET necessary for the proper activity of PETase enzymes [138,139]. The most advanced enzymatic PET degradation is carried out by Carbios company in French which collaborates with the most significant industrial enzyme production. Novozymes company in Danish is going to commercialize this PET enzymatic recycling process.

The decomposition of PET recycling by lipase enzyme as a valuable biocatalyst comprised of (i) hydrolysis of BHET and (ii) esterification of the resulted products (mono (2-hydroxyethyl)terephthalic acid, MHET and TPA) using dimethyl carbonate [140]. The degradation of PET nanoparticles prepared from PET films as substrate by TfCut2, a polyester hydrolase from *Thermobifida fusca*, was studied in the presence of EG, TPA, BHET and MHET. The initial reaction rates were determined and kinetically analyzed using a Michaelis–Menten reaction kinetics model [141].

PET biodegradation currently one of the popular methods to degrade PET. This process’s advantages are environmentally friendly and low cost, e.g., 3 Kg enzyme can degrade 1000 Kg of PET with the cost of up to 63 €. Meanwhile, the byproduct of that process can be applied to different applications. However, biodegradation’s reaction time took longer than that by chemical and mechanical techniques [6,24,142]. Several kinds of bacteria and fungi were discovered to depolymerize PET into short chains of oligomer and monomers (BHET and MHET) [143]. The *Ideonella sakaiensis* bacteria was discovered by Yoshida et al., This kind of bacteria proved to degrade PET effectively [144]. In addition, Esterase is a member of the enzyme that could be cut the ester bond (short-chain alkyl ester). That kind of bacteria could be discovered at surface modification of PET. Hence, the initial degradation of PET took place by *bacillus* and *nocardia* via esterases [145].

The summary of enzymatic degradation of polyethylene terephthalate with a different microorganism type is shown in Table 3. Furthermore, *Cutinases* is one of the enzymes that demonstrated the capability of degradation of hydrophobic PET. The *cutinases* are found from the plant with viral pathogenic fungi, whereas; it promotes the fungal penetration to hydrolyzed *cutin.* The form of *cutin* is produced from oxygenated C-linked cross-linking with the range temperature of oxygenated 16–18 °C. In addition, the ester chain of linking is hydrolyzed by *cutinases* activity with endo and exotherm hydrolyzed process [146]. Furthermore, Ribitsch et al. investigated the *cutinase* hydrophobic fusion protein with HFB4, HFB7 and HFB9b shown without GST fusion. It could stimulate PET hydrolysis by cutinase 1 from *T. cellulosilytica* [147]. Furthermore, the biological valorization of PET monomers was studied by Kim et al. The first step of the process was rPET could be degraded by chemically hydrolysis process became TPA and EG. TPA could be converted to gallic, vanilla, mumocic acid and pyrogallol by the process of funneling intermediate protocatechuic acid (PTA) with metabolically engineered microbes. In addition, the product of glycolic acid obtained from EG–fermenting process of microbe. DCD 1,2 DCD, 1,2-di hydroxy-3,5-cyclo hexa diene-1,4-di carboxylate [148].

### 6.6. PET Recycling Using Methanolysis–Hydrolysis

The first stage in the methanolysis–hydrolysis process is reacted with superheated methanol vapor to get DMT, monomethyl terephthalate, EG oligomeric products. The next step is to remove residue with mainly containing oligomers with fractional distillation after TPA’s precipitation. The reaction was completed after 15 min, which leads to a significant decrease in the time required for the degradation of the polymer process. On the other hand, conventional hydrolysis needs to take 45 min. The Mitsubishi heavy industries Ltd. established the latest technology of methanolysis–hydrolysis with plant scale. Figure 7. shows that the process of diagram recycling PET by methanolysis–hydrolysis of Mitsubishi process. The PET begins with collecting the waste of PET, and the polyethylene goes to the shredding unit. The function of the shredding unit is to get a small size of particles of PET. The PET through the depolymerization unit with added methanol and the following processes is divided into two processes: DMT purification process and EG/methanol purification. The next step of DMT purification must be passed to the hydrolysis unit to become a TPA product, and it will become the raw material of PET [162,163,164,165,166].

### 6.7. PET Recycling Using Glycolysis–Methanolysis

The recycling of PET with a combination of glycolysis–methanolysis starts from dissolving PET in a mixture of EG, TPA, DMT and oligomers. The reactant will be reacted at the superheated methanol condition with the reaction temperature range of 250–290 °C to get a large amount of DMT. Furthermore, the hybrid process technology to recycle by glycolysis–methanolysis is developed to get the high yield of product and production rate. The different types of reaction parameters were designed with the methanolysis process, which had a better role than the glycolysis process to increase the product yield with the ratio of EG and PET = 0.52 and the temperature reaction 240 °C [167,168].

### 6.8. PET Recycling Using Glycolysis–Hydrolysis

The combined application of glycolysis–hydrolysis had been discovered in literature since 1986. The patent claimed that the recycling of PET by extruder with increasing reaction temperature up to 280 °C with the slightest amount of EG before hydrolysis, the molecular weight would be reduced from 30,000 to 9000–1000. Furthermore, it depends on the quantity of EG. Moreover, the reaction time would reduce from 45 to 12–15 min to get EG and TPA. The high reaction of temperature and a large excess of reactants was considered to get many conversions and achieve low–intermediate molecular weight. It considers that hydrolysis and glycolysis are reversible reactions, and the equilibrium of polymerization with the reverse reaction of polyesterifications must be changed [169,170,171]. Figure 8 shows the possible reaction PET by the glycolysis–hydrolysis process. Optimizing the economic process with a capacity of a plant of 8000 t/year, the unitary cost of producing 1 kg of BHET and TPA are 1.99 €/kg BHET and 1.02 €/kgTPA, respectively [172].

### 6.9. PET Recycling Using Steam Hydrolysis

The recycling of PET using steam hydrolysis is carried out under high temperature and pressure. This process can be applied to a continuous process with the temperature range of was 200–300 °C under high-pressure steam up to 15 atm. The super-high steam is conducted at the bottom of the hydrolysis zone following the condensation process [173]. Rosen et al. studied the steam hydrolysis of PET with a mechanical process. The PET waste was crushing or grinding. Furthermore, it removed the impurities of waste PET by water. It was heated by increasing the temperature range between 221–316 °C to hydrolyze the solution and then obtained crude TPA in a cooling process [174]. PET waste was decomposed to main product TPA (90%) in sub- and supercritical water at short reaction times (1–30 min) in a batch reactor at temperatures from 250 to 400 °C by Čolnik et al. [175], which by-product was benzoic acid, 1,4-dioxane, acetaldehyde, isophthalic acid and CO_2_.

### 6.10. PET Recycling Using Solid-State Hydrolysis

Strukil studied the solid-state PET hydrolysis by mechanochemical milling and vapor-assisted aging [138]. Mechanochemical PET hydrolysis does not depend on the properties of plastic substrate such as crystallinity and uses low energy. Benzaria et al. presented that the PET hydrolysis was conducted in the reactor-mixer-extruder communicate with a sealed heating chamber at a temperature between 50 °C and 200 °C for a time sufficient to complete the saponification, for example, 5 to 30 min [176].

## 7. Factor for Recycling PET

### 7.1. Crystllility of PET

The physical properties of PET are one of the crucial parts for recycling PET due to PET had a low crystallization rate. A method to increase crystallization rate with heat up of the polymer became melt and incorporation of crystallization facilitator and nucleating agents [177]. The crystallization rate of PET between 150–180 °C depends on the degree of chain orientation, molecular weight and the polymerization catalyst’s nature. It is applied in the process production of PET [178,179]. The crystallinity is induced by heat up until the glass transition temperature (Tg) is reached and followed by a molecular orientation. Furthermore, getting 100% crystallinity under the lowest free energy is not possible due to the polymer’s molecular weight didn’t have an unvaried form. Some of the polymers could react half part of the reaction semicrystalline [180]. At the hydrolysis process, the powerful effect is PET crystallinity due to the crystallites perform as barriers between moisture and oxygen diffusion [181].

Wan et al. reported that after the reactions, the residual solids of PET almost remained in flake shape, and their molecular weights were close to that of PET before the reaction [182]. Kint et al. reported that the PET degradation decreased the melting enthalpy of fully crystalline PET, increasing the degree of PET crystallinity in the range of 34% to 49% [183]. The crystal structure of post-consumed PET by catalytic glycolysis with novel mesoporous metal oxide spinel catalyst had been studied by Imran et al., The catalyst like ZnO (hexagonal), mixed metal oxide spinel (ZnMn_2_O_4_, CoMn_2_O_4_ and ZnCo_2_O_4_) and metal oxide spinel (CO_3_O_4_ and Mn_3_O_4_). It is shown that the yield of BHET would be increased up to 92.2 mol% with zinc manganite tetragonal spinel (ZnMn_2_O_4_) under pressure, and the temperature of the reactions were 260 °C and 5.0 atm, respectively. The tetrahedral Zn^+2^ ion and octahedral Mn^+3^ ion collaborates with the spinel crystal structure [184]. Hence, the high crystallinity of PET is not readily degraded in PET degradation.

### 7.2. Particle Size of PET

The conventional methods to reduce the extremely small particle size of PET are used by using ball mills, grinding process, cryogenic grinding, pulverizing, attrition mills and other conventional methods. Nevertheless, there wasn’t a method effective to reduce the size particle of rPET from single used on bottle became small on particle size up to 500 microns. The process to produce extremely small recycled polyethylene terephthalate particles are maintain the temperature over the temperature of glass transition temperature (Tg), whereas the flakes RPET is over 150 °C. The flakes are held at an elevated temperature several times. Furthermore, the crystallinity of rPET is increased slightly, and it could be a subject of a separate process with RPET flakes that had a greater particle density [185].

Ügdüler et al. reported that the experiments performed at optimal conditions with different post-consumer plastic waste types revealed that the degradation rate increases inversely proportional to the particle size. Furthermore, the increased thickness of the samples and the presence of multilayers reduce the decomposition yield by a factor of two, as observed for monolayer (80%) versus multilayer PET trays (45%). Using the optimized alkaline hydrolysis with further cleaning processes, the different types of colors, including carbon black, were removed from the hydrolysate successfully [186]. Mancini and Zanin reported that the PET hydrolysis occurred preferentially at the chain ends and superficially, having as controller mechanism the acid diffusion into the polymer structure [187]. The waste of PET reduce into small particles could be used as part of filler in the adhesive formulation [11,188]. Mehrabzadeh studied the particle size effect on the PET hydrolysis process with the temperature reaction 130–170 °C and 7 M H_2_SO_4_ with a reaction time of 5 h. The particle size would be reduced, and it would be affected by the increasing yield depolymerization of PET. Furthermore, the fine particles were the most significant particle type to obtain the highest yield up to 100% with the size of particle 0.2–0.25 mm [189].

### 7.3. Effect of Amount of Catalyst

The amount and type of catalyst would affect the depolymerization of PET. Chen et al. investigated the quantity of Zn/Al as a catalyst to affect the yield of BHET. Furthermore, a few PET could be degraded in the absence of the catalyst. On the other hand, the rising ratio of catalyst to PET from 0% to 0.5% would dramatically increase the yield of BHET from 3.2% to 53.2%, with the activation energy up to 79.3 kJ/mol [190]. The kinetic rate constant as a function of the ratio of PET to the catalyst was studied by Lopez et al. For the glycolysis process with Na_2_CO_3_ as the catalyst and the temperature reaction up to 196 °C, the rate constants for the molar ratios of PET to catalyst (100:1, 200:1 and 400:1) were 67.2, 1.57 and 0.858 h^−1^, respectively [101]. On the other hand, the important parameters of glycolysis conversion was time < temperature < amount of catalyst with the major products were BHET and oligomers [191].

### 7.4. Downstream Treatment

#### 7.4.1. After Hydrolysis

After crystallization of NaTPA, the aqueous filtrate is charged into electrodialyzer using a bipolar membrane to individually separate acid and base of alkali metal/earth metal hydroxide. The acid so separated and recovered is fed back to the neutralization tank, while alkali metal hydroxide/earth metal is also fed back to the reaction tank. Alkali metal/earth metal salt (e.g., sodium hydrochloride or sodium sulfate) contained in the said filtrate is electrolyzed in a more detailed manner. Then, alkali metal/earth metal ion such as Na^+^ ion passed through cation membrane, is bound with OH^−^ ion to form alkali metal/earth metal hydroxide such as sodium hydroxide. Additionally, the acidic anion such as Cl^−^ ion or SO_4_^2−^ ion, passed through anion membrane, is bound with H^+^ ion to form acids such as hydrochloric acid or sulfuric acid. The slurry of TPA, so obtained from said neutralization process, is charged into a crystallization tank to sufficiently enlarge the particle size of TPA. Crystallization temperature drop in the tank should be preferably within 30 to 50 °C [192]. After crystallization, the recovery of TPA was a nanospindle-shaped crystal [193], but this size is smaller than that of typical TPA size (100 μm).

#### 7.4.2. After Glycolysis

Goh et al. reported that BHET produced from the PET waste glycolysis process was purified and compared using two stages evaporation or crystallization processes [109]. The optimum conditions of 3 h crystallization time, 2 °C crystallization temperature and 5:1 mass ratio of distilled water used to glycolyze solid gave the highest yield and purity of the crystallization process [108].

### 7.5. Complex of Plastics

The change made it easier for the company to remove the dyes, pigments, foreign polymers and even ketchup and mayonnaise that can end up in bales of waste PET. The material will contain, on average, 15% contamination [45]. The most used plastics were low-density polyethylene (LDPE), high-density polyethylene (HDPE), polyethylene terephthalate (PET), polypropylene (PP) and polystyrene (PS), better known as the ‘‘big five’’. Aguado et al. studied the PET complex as feedstock was conducted by glycolysis and hydrolysis [172]. Since oligomers are not included in chemical databases, their identification is a complex process. The analyses were carried out by UPLC-MS-QTOF. The use of high-resolution mass spectrometry allowed the structural elucidation of these compounds and their correct identification [1]. Some material is too complex or high impurity to separate PET. Pyrolysis should be used to decomposed this material. Pyrolysis cracks long polymer chains into short-chain hydrocarbons like diesel and naphtha under low-oxygen conditions and temperatures of more than 400 °C.

## 8. Kinetic and Modeling Analysis of Chemical Recycling PET

For the kinetic degradation of PET by thermal degradation, generally, there are two different temperatures to investigate thermal degradation for instances, 480 and 780 °C. Based on thermogravimetric analysis (TGA), the thermal degradation of PET starts at 410 °C and 470 °C with a heat rate of 10 K/min. Furthermore, the char product occurs as the temperature degradation process higher than >600 °C [194]. A comprehensive analysis has been performed regarding the kinetic model. The different types of kinetic models were examined considering the rate-limiting diffusion, nucleation or the reaction itself. The kinetic parameters and thermal behavior of depolymerization were determined by isoconversional method reported by Friedman [195]. The polymer sample decomposition was calculate based on Equation (1) [196]:(1)$$\frac{d\alpha }{dt}=kf\left(\alpha \right)$$
(2)$$\frac{d\alpha }{dt}=A\exp\left(-\frac{E}{RT}\right)\;f\left(\alpha \right)$$
*f*(*α*) is the reaction model, *T* is absolute temperature, *A* is a pre-exponential factor (s^−1^), E is the activation energy (kJ/kmol) and R is the universal gas constant (kJ K^−1^·kmol^−1^). The degree of conversion (α) is defined and expressed:(3)$$\alpha =\frac{m_{o}-m_{t}}{m_{o}-m_{f}}$$
where *m_o_*, *m_t_* and *m_f_* are the initial mass, mass at the time and final mass during lignin pyrolysis. All isoconversional methods have their origin in the isoconversional principle, which states that the reaction rate at the constant extent of conversion is only a function of temperature [197]. Equation (3) is taken nature log to be given:(4)$$\ln\left(\frac{d\alpha }{dt}\right)=\ln\left(A.f\left(\alpha \right)\right)-\frac{E}{RT}$$

The apparent activation energy can be obtained from the slope (E/*R*) of the linear plot of ln (*dα*/*dt*) vs. 1/*T* for each value of conversion *α*. Some results are shown in Table 4. Coney, et al., studied the kinetic degradation of PET by Kissingers, Freeman–Carroll, Chatterjee-Conrad and Friedman method with a rate of air by thermogravimetric analysis °C/min with the activation energy up to 202.1, 160.2–411.5, 14.0–301.4 and 69.6–210.8 kJ/mol, respectively [198]. On the other hand, Jenekhe et al. studied PET degradation by nitrogen gas with the range of rate nitrogen gas up to 0.31–10 °C. The activation energies by the Arrhenius plot and iso–conventional method was 198.6 and 173.6, respectively [199].

Chen et al. presented that the conversion of PET fiber reached about 25.5% only 5 min. Fiber breakage can be observed after the reaction time of 8 min. The conversion of PET reached the maximum value after 15 min, but the yield of BHET reached a plateau value representative of reaction equilibrium after 20 min. This result indicates that equilibrium should be reached [190]. After 25 min, the viscosity-average molecular weight was lower than 800. The kinetic models of PET glycolysis and hydrolysis are followed Equations (5)–(7) calculated the PET conversion and TPA yield:(5)$$X_{PET}=\frac{W_{PET,0}-W_{PET,f}}{W_{PET,0}}$$
(6)$$Y_{TPA}=\frac{W_{TPA}/M_{TPA}}{(W_{PET,0-W_{PET,f})/M_{PET}}}$$
(7)$$Y_{BHET}=\frac{W_{BHET}/M_{BHET}}{(W_{PET,0-W_{PET,f})/M_{PET}}}$$
where *W_PET,0_* = initial weight of PET (g); *W_PET,f_* = residual weight of PET (g); *M_PET_* = molecular weight of PET repeat unit (g); *W_TPA_* = weight of TPA obtained by hydrolysis (g); *M_TPA_* = molecular weight of TPA; *W_BHET_* = weight of BHET obtained by hydrolysis (g); *M_BHET_* = molecular weight of BHET.

The PET depolymerization is just considered the PET solid in the solution. The reaction mechanism is:(8)$$\mathrm{PET}\overset{{\;k}_{1}}{\to }\;\mathrm{monomer}$$

The rate expression for a pseudo-first-order kinetic of PET depolymerization is by following:(9)$$\frac{dX_{PET}}{dt}=k\left(1-X_{PET}\right)$$
where *k* is the forward reaction rate constant. Furthermore, the reaction rate constant is defined by the Arrhenius equation:(10)$$k=k_{o}exp\left(\frac{-Ea}{RT}\right)$$

The rate expression for a pseudo-second-order kinetic of PET irreversible hydrolysis of PET and base salt is [182]:(11)$$-\frac{dC_{PET}}{dt}=kC_{PET}C_{base}$$

The PET reversible glycolysis for pseudo-second-order kinetics is [101,200]:(12) PET+EG →very fast oligomers →fast dimer  ← k ′→kBHET

The rate expression for PET glycolysis is given
(13)$$-\frac{dC_{PET}}{dt}=kC_{PET}C_{EG}C_{cat}-k\prime C_{BHET}C_{cat}$$

The PET reversible hydrolysis for a pseudo-second-order kinetics is:(14) PET+H2O  Na-base→very fast  oligomers →fast dimer  ← k ′→kNa2PTA+EG

The rate expression for PET hydrolysis is given:(15)$$-\frac{dC_{PET}}{dt}=kC_{PET}C_{EG}C_{cat}-k\prime C_{Na_{2}TPA}C_{EG}C_{cat}$$

The equilibrium constant is shown as:(16)$$K=\frac{k}{k^{\prime }}=\frac{C_{TPA}C_{EG}C_{cat}}{C_{PETC_{EG}}C_{cat}}$$
where *k′* is the backward reaction rate constant, and *C_cat_* is the concentration of catalyst.

Currently, hydroxyl number is used to measure BHET and oligomer’s concentration and discuss the kinetic of PET glycolysis [201,202]. The BHET and dimer are only obtained after reaction when the mole ratio of EG/PET is large. The oligomer products were discussed [201]. The kinetic of PET with pyrolysis in a spouted bed reactor was studied by Niksiar et al. The temperature range of kinetic pyrolysis PET was 450–560 °C using a different type of particle size (0.1–1.0 mm and 1.0–3.0 mm) with the activation energy of 276.8 and 264.3 kJ/mol [203]. The reaction kinetic of PET by Arrhenius law also was studied by Brems et al., the value of activation energy is 237 kJ/mol while the pre-exponential factor (A) up to 10^16^ s^−1^ at a heating rate (β) = 3 °C·min^−1^ to 2.5 × 10^16^ s^−1^ at β > 100 °C min^-1^ [194].

Table 4 lists the kinetic and modeling analysis of PET with a different type of condition operation. The hydrolysis of PET with nitric acid by the shrinking core model with surface chemical reaction as a rate-controlling step was investigated by Kumar et al. to find the kinetic parameters and the activation energy for the chemical reaction up to 135 kJ/mol [204]. Chen et al. already investigated the kinetic glycolysis of PET with zinc catalyst was examined in a pressurized reactor. The activation energy of PET without a catalyst is 108 kJ mol^−1^. On the other hand, the activation energy with zinc acetate will be decreased up to 85 kJ mol^−1^ [205].

In PET methanolysis, the BHET degradation reaction in supercritical methanol was a first-order reaction. This series reaction is given:(17)$$BHET\overset{\;k_{1}}{\to }\;MHET\;\overset{\;k_{2}}{\to }\;DMT$$

The reaction rate expression is
(18)$$-\frac{dC_{MHET}}{dt}=k_{1}C_{BHET}-k_{2}C_{MHET}$$
(19)$$-\frac{dC_{DMT}}{dt}=k_{2}C_{MHET}$$

Genta et al. presented the *k*_1_ and *k*_2_ values for PET-oligomer, BHET and MHET by methanolysis [206]. Goje and Mishra reported the effect of temperature on glycolysis rate constant (*k*), condensation rate constant (*k′*), equilibrium constant (*K*), Gibbs free energy (ΔG), enthalpy (ΔH) and entropy (ΔS) of depolymerization of PET processed with EG (40 mL) for 60 min reaction time; for PET particle size of 127.5 mm; using 0.002 mol of zinc acetate catalyst at 1 atm pressure conditions [207].

The modeling and PET analysis are still highlighted to get the best condition to degrade PET into a potential small molecular compound. Farzia et al. studied the kinetic modeling of PET by biodegradation process with Michaelis–Menten activation or inhibition model [208]. The inhibition model was applied, whereas the reactant concentration will be raised slightly, affecting a decrease in the reaction rate. However, in the case of the activation model, it increases the reaction rate. The Michaelis–Menten inhibition model is given as follows:(20)$$-r_{A}=\frac{kC_{A}^{2}}{1+K_{1}C_{A}+K_{2}C_{A}^{2}}$$

Integration with a first-order equation:(21)$$\frac{1}{C_{AO}\left(1-X\right)}-k_{1}\log\;\left(1-X\right)+k_{2}C_{AO}X=\mathrm{kt}-1$$

The depolymerization of PET powder was studied by Farzi et al., the kinetic models to predict the bio-degradation of PET [208]. Base on the activation energy in Table 4, the activation energy is low when the catalyst or ionic liquid is used. However, they decrease with increasing reaction temperature.

## 9. Reactor of Recycling PET and Application of rPET

### 9.1. Reactor and Product Distribution

The main objective of recycling PET is to convert the feedstock and treat from different recycling methods to get high-end value products and decrease consumption energy for different kinds of reactors. It obtained different product distribution. Table 5 shows that the various types of the reactor of recycling PET with product distribution. Bai et al. studied the gasification process of PET by supercritical water with quartz tube reactor, the high temperature for the gasification processes up to 700–800 °C. The common monomer product is biphenyl with H_2_ and CH_4_ with the conversion rate of carbon (CE) within 98.00 wt% at 800 °C. On the other hand, the temperature is decreased at 500 °C. The major products are CO, CO_2_, benzoic acid, etc. Furthermore, the effect of seawater components on the gasification process has also been studied. The saltwater was a capacity utilization. The route to plastic gasification and the metal materials inside saltwater promoted PET gasification in different ways [220].

The hydrolysis process of PET under microwave assisted was explored by Liu et al. The depolymerization process was completed with the pressure up to 200 bar with the range temperature reaction up to 90 – 120 min and the weight ratio between water / PET was 10:1, the majority of products are EG and DG [231]. The PET hydrolysis process by stainless steel high–pressure reactor was studied by Valh et al., and the maximum conversion of TPA was 92%. The temperature and pressure were used up to 250 °C and 39–40 bar, respectively [232].

### 9.2. Application of Product after PET Upcycling

Weakabayashi studied the repolymerization of TPA and EG from rPET under the hydrothermal condition in the semi-batch reactor. The percentage of total yield 96.2% on carbon weight, with temperature reaction of aminolysis at 473 K over 50 min. The yield of TPA and EG is closed to theoretical ones. Other amines as solvent (methylamine, ethylamine, dimethylamine, trimethylamine) have less intermediate product yields [92]. Pulido et al. prepared the porous membranes of rPET by nonsolvent-induced phase separation, and evaluated for ultrafiltration with a molecular weight cutoff of 40 kg·mol^−1^ in dimethylformamide at temperatures up to 100 °C [53].

Lee et al. studied that the glycolytic products BHET from rPET were reacted with adipic acid to yield polyester polyols. The polyester polyols were then reacted with either MDI or TDI to obtain polyurethanes [233]. Yang et al. synthesized a series of thermoplastic polyester elastomer and thermoplastic poly(ester amide)s elastomer) copolymers were obtained by depolymerizing PET [234]. Chen et al. synthesized polyamide 66 (PA66) copolymers with aromatic moieties from rPET [235]. PA66/(BAHT-AA) copolymers were synthesized. AA is adipic acid. Bis(6-aminohexyl) terephthalamide (BAHT) is synthesized by the aminolysis reaction between BHET from rPET and hexamethylenediamine.

Tiso et al. presented the sequential conversion of rPET into two types of bioplastics: a medium chain-length polyhydroxyalkanoate (PHA) and a novel bio-based poly(amide urethane) (bio-PU) [157]. PET films were hydrolyzed by a thermostable polyester hydrolase yielding 100% terephthalate and ethylene glycol. A terephthalate-degrading *Pseudomonas* was evolved to also metabolize ethylene glycol and subsequently produced PHA. The strain was further modified to secrete hydroxyalkanoyloxy-alkanoates, which were used as monomers for the chemo-catalytic synthesis of bio-PU.

Rorre et al. developed an upcycled PET to higher-value, long-lifetime materials, namely fiber-reinforced plastics (FRPs), via combination with renewably sourceable monomers (olefinic diacids and olefinic monoacid). By harnessing the embodied energy in reclaimed PET and implementing renewably sourceable monomers with specific chemical functionality relative to petroleum building blocks, the resultant rPET-FRPs exhibit considerably improved material properties and are predicted to save 57% in the total supply chain energy and reduce greenhouse gas emissions by 40% over standard petroleum-based FRPs [137], which process cost was discussed too. Mirjalili et al. upcycled rPET to a microporous carbon structure that functions as a double-layer supercapacitor substance for energy storage [54].

Ashoor et al. presented that rPET was added to asphalt by a percentage of (5–15%) to improve its properties of mechanical and rheological properties [236]. Sposito et al. reported incorporating rPET wastes in rendering mortars based on Portland cement/hydrated lime [237]. In this case, rPET fibers were proposed to be used as either reinforcement in concretes or cast as blocks, which can be accepted as successful building materials. Even though PET fiber reinforced concrete offers less compression strength and flexural rigidity than conventional concrete, it offers high ductility, thereby increasing the concrete’s deforming capability [13]. Jung et al. synthesized the magnetic porous carbon composite derived from metal-organic framework using rTPA from PET waste bottles as organic ligand and its potential as adsorbent for antibiotic tetracycline hydrochloride [238]. rPTA can be applied as a synthesis of metal-organic framework materials [239] and as a dual-acid catalyst to produce furfural from xylose in a water-toluene biphasic system [240].

## 10. Conclusions

There are considerable challenges to recycling PET not only from the process but also in the type of PET waste. The water, coloring, acetaldehyde and heavy metal contamination are affected on the recycling process of PET. Furthermore, the degradation of PET had been developing for several methods and technologies, for instance, mechanical and chemical recycling. For mechanical recycling, some recycling materials are downcycling material or limited with recycle time. Except that, rPET materials and articles produced by modern super-clean technologies can be considered safe indirect food applications in the same way as virgin food-grade PET because rPET contains the impurity (acetaldehyde).

The recycling of PET by chemical process had so many methods such as degradation, hydrolysis process by alkaline and acid hydrolysis, methanolysis, glycolysis, microwave, ionic liquid, phase–transfer catalysis, combination glycolysis-methanolysis, glycolysis–hydrolysis, methanolysis–hydrolysis. According to the kinetic result, the reactivity of PET recycling is evident and easy to design the reaction system. The different types of reactor and chemical depolymerization will be affected to the product distribution, and the common monomer products are BHET, DMT and TPA. The glycolysis process is a promising method for PET recycling due to the highest yield of BHET. The application of BHET is used in the large scale of petrochemical industry. However, the challenge for the company is to remove the dyes, pigments, foreign polymers, which will contain average 15% contamination before PET chemical recycling. It is crucial to develop a separation method to isolate PET from PET waste. On the other hand, the purification of TPA to obtain a large particle size (100 μm) is still required to investigate.

## Figures and Tables

**Figure 1 polymers-13-01475-f001:**
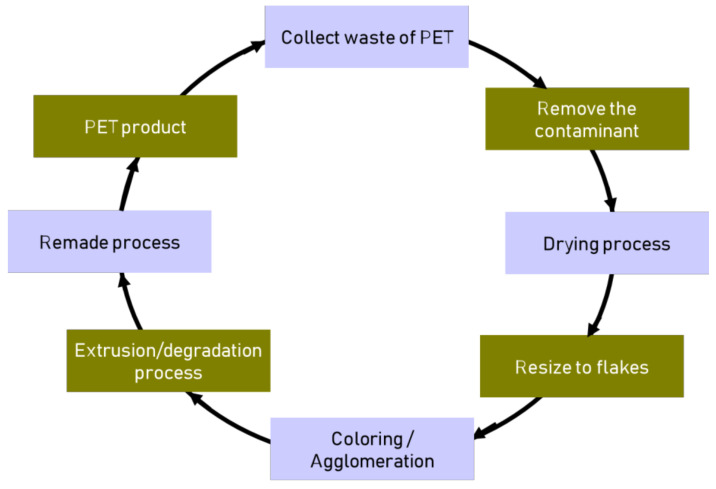
Scheme of the conventional mechanical process of PET recycling.

**Figure 2 polymers-13-01475-f002:**
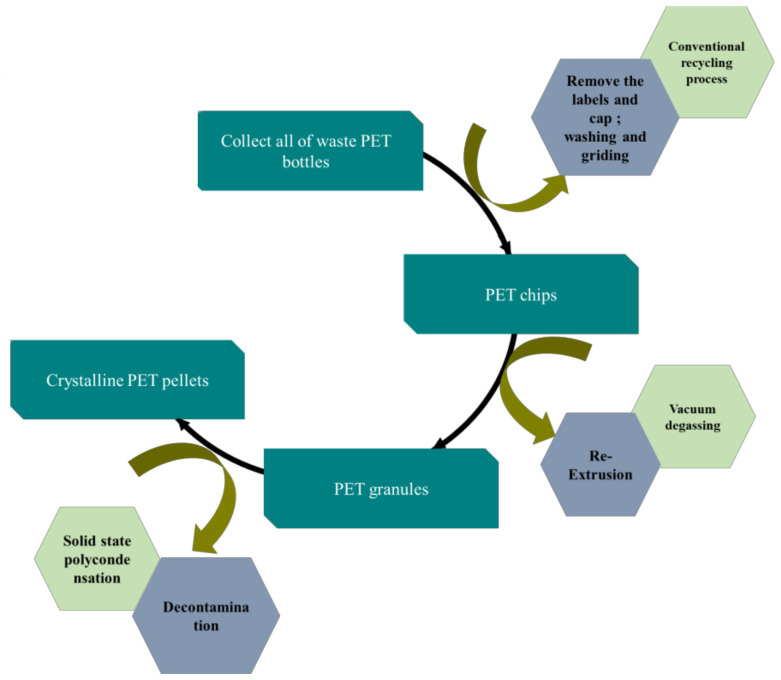
Scheme of PET super-clean recycling processes based on bottle–chip–granule–pellet, and mechanical methods.

**Figure 3 polymers-13-01475-f003:**
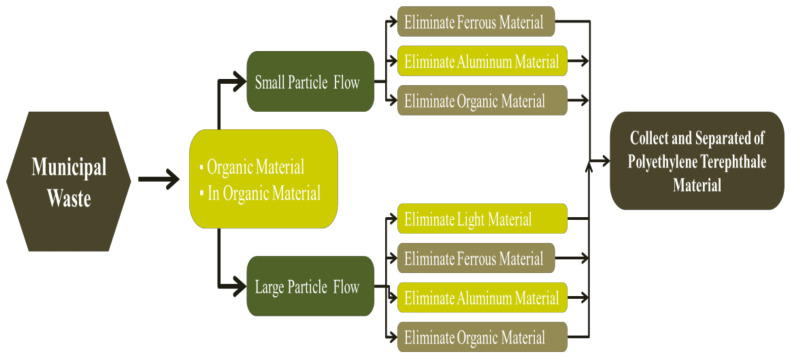
The classification of the conventional process to screen ferrous, aluminum and organic materials and collect PET.

**Figure 4 polymers-13-01475-f004:**
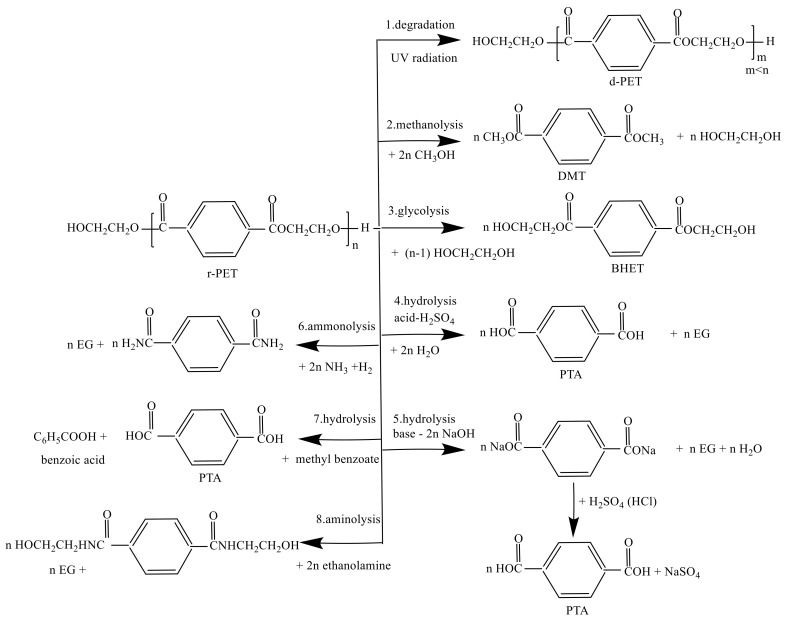
The 8 possible reactions of recycling PET including methanolysis, glycolysis, hydrolysis, ammonolysis and aminolysis.

**Figure 5 polymers-13-01475-f005:**
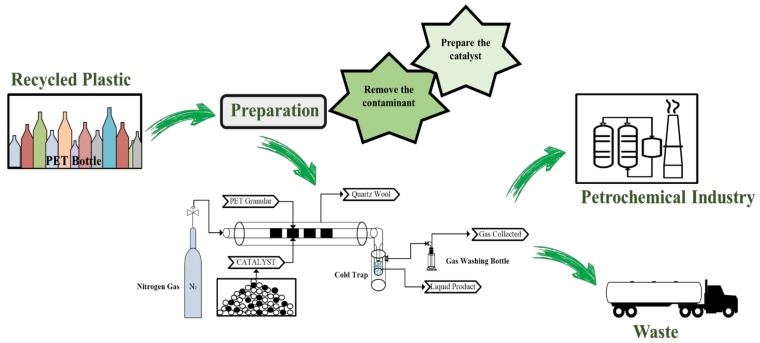
Schematic of PET pyrolysis by a tube reactor.

**Figure 6 polymers-13-01475-f006:**
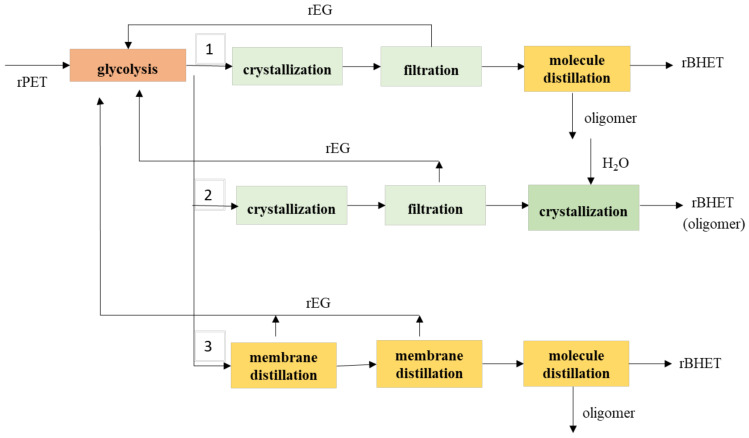
The 3 possible process path of PET glycolysis with different separation methods.

**Figure 7 polymers-13-01475-f007:**
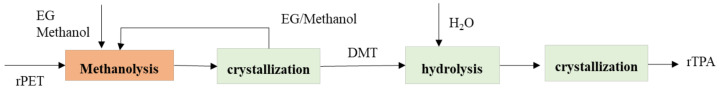
The process of PET recycling with methanolysis–crystallization-hydrolysis.

**Figure 8 polymers-13-01475-f008:**
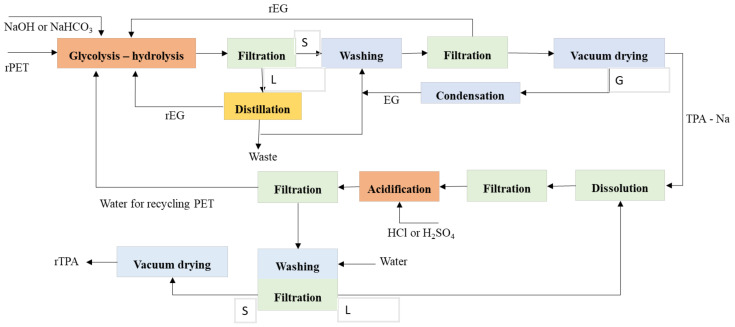
The possible reaction of PET by glycolysis–hydrolysis with the separation process.

**Table 1 polymers-13-01475-t001:** Industrial rPET production and their locations.

Company	Reaction	Country/Region	Ref
FENC	hydrolysis	Taiwan	[58]
Gr3n	hydrolysis	Switzerland	[59]
JEPLAN (PRT)	glycolysis	Japan	[60]
Garbo	glycolysis	Italy	[61]
IFPEN	glycolysis	France	[62]
Ioniqa	glycolysis	Netherlands	[63]
PerPETual	glycolysis	UK	[64]
Poseidon Plastics	glycolysis	UK	[65]
Eastman	methanolysis	USA	[66]
Loop industries	methanolysis	Canada	[67]
DePoly		Switzerland	[68]
Carbios	enzyme	French	[69]
Agiylx	pyrolysis	USA	[45]
Pyrowave	microwave radiation	USA	[45]

**Table 2 polymers-13-01475-t002:** The yield of BHET from PET glycolysis with different kinds of catalysts.

Type of Catalyst	Reactor	Time of Reaction, min	Molar Ratio Catalyst to PET	Particle Size	Temperature °C	Yield_BHET_ %	Ref
Na_2_CO_3_	Batch	60	1:100	0.25 mm	196	78.8	[101]
Na_2_SO_4_	Batch	60	1:380	0.25 mm	196	19.6	[101]
NaHCO_3_	Batch	60	1:190	0.25 mm	196	60.2	[101]
Titanium (IV)-phosphate with EG	Necked flask	150	0.3:0.13	NA	190–200	97.5	[102]
Zn(OOCCH_3_)_2_ with EG	Necked flask	150	0.3:0.13	NA	190–200	62.8	[102]
di-n-propylamine with EG	Necked flask	90	1:4	NA	160	73.7	[103]
di-n-butylamine with EG	Necked flask	90	1:4	NA	160	76.8	[103]
N-propylamine with EG	Necked flask	90	1:4	NA	160	72.8	[103]
diisopropylamine	Necked flask	90	1:4	NA	160	70.1	[103]
γ-Fe_2_O_3_nanoparticles	NA	60–80	1:8 (wt%)	>50 µm–<10 µm	255–300	80–90	[104]
Fe_3_O_4_-boosted MWCNT	NA	120	1:19 (wt%)	>50 µm–<10 µm	190	100	[104]
Mg-Al-O@Fe_3_O_4_	NA	90	1:199 (wt%)	>50 µm–<10 µm	240	80	[104]
Zinc Acetete	NA	480	1:99 (wt%)	>50 µm–<10 µm	198	75	[104]

**Table 3 polymers-13-01475-t003:** Summary enzymatic degradation of PET with a different type of microorganism.

Type of Enzyme	Microorganism	Materials	Temperature (°C)	Product	Ref
Cbotu_EstA	*Clostridium botulinum ATCC3502*	PET film	50	TPA,MHET	[149]
Lipases	*Candida cylindracea*	PET nanoparticels	40	1,2-Ethandiol, TPA	[150]
Tcur0390	*Thermomonospora curvata DSM43183*	PET nanoparticels	50	NA	[151]
Hydrolase TfH	*Thermobifida fusca DSM43793*	PET pellete	44–55	TPA, EG	[152]
TfCut1	*Thermobifida fusca KW3*	PET nanospheres	55–65	HEB, MHET, BHET, Benzoic acid	[153]
Tha_Cut1	*Thermobifida alba DSM43185*	PET	50	TPA, Benzoic acid, HEB, MHET	[154]
MHETase and PETase	*Ideonella sakaiensis*	PET	30	MHET	[154]
BsEstB	*Bacillus subtilis 4P3-11*	PET	40–45	TPA, Benzoic acid, MHET	[155]
LCC cutinase	*Leaf-branch compost*	Amorphous PET	50–70	MHET, TPA, Benzoic acid	[156]
Polymer polyhydroxyalkanoate	*Pseudomonas*	PET	70	EG,TPA	[157]
Hydroxyalkanoyloxy-alkanoate	*Pseudomonas*	PET	70	EG,TPA	[157]
Thc_Cut2	*T. cellulosilytica*	PET films	50	MHET, TPA, BA, HEB	[158]
Arg29Asn	*Mutant The_Cut2*	PET films	50	MHET, TPA, BA, HEB	[158]
Lipases	*Aspergillus oryzae CCUG 33812*	PET fabric	55	BHET,TPA	[159]
Gln65Glu	*Mutant The_Cut2*	PET films	50	MHET, TPA, BA, HEB	[158]
TfCut2	*Thermobifida fusca KW3*	PET films	60	MHET	[160]
Thh_Est-esterase	*Thermobifida halotolerans*	3PET model substrate	50	MHET,TPA, BA, HEB	[161]

**Table 4 polymers-13-01475-t004:** Kinetic and modeling analysis of PET with different types of condition operation.

PET Catalyst		Method	T (°C)	k	k_2_	K	A	Ea (Kj·mol^−1^)	Y_BHET_, %	Y_DMT_, %	Y_TPA_, %	Ref
PET		P	373–443 448–503				6.4 × 10^25^/min 9.31 × 10^11^/min	347.4 172.6				[73]
PET		P	390				48 × 10^21^/min	323.8				[209,210]
PET		P						202.1				[198]
PET		P						198.6				[199]
PET	H_2_O	H	180	0.0037				378				[208,211]
flake	H_2_O	H	265	0.352 g PET/(mol min)		0.664		55.7				[200]
PET	H_2_@ZSM-5-25	H	230	0.1787				1.2			98.5	[212]
PET	H_2_O without catalyst	H	230	0.0755				19.4				[212]
flakes	NaOH	H	200	0.1900				99			97.9	[95]
flakes	KOH	H	160	2.24 × 10^−6^ L/(min cm^2^)			419 L/(min cm^2^)	69			90.9	[182]
PET	HNO_3_	H	100	0.4800				101.3				[213]
PET	HNO_3_	H	88	1.9 × 10^−2^ L/(min cm^2^)		0.042	659 L/(min cm^2^)	88.3				[214]
Flake < 2	sulfuric acid	H		2.0 × 10^11^ g.cm/(mol min)				99.7			90	[187]
PET	TBAB	Hydrolysis	60–80	0.1100				75			99.0	[215]
flake	3Bu6DPB, NaOH	H	90	0.2 L/(mol h)			1.3 × 10^9^ L/(mol h)	68.2			>90	[134]
flake < 6	TOMAB NaOH	H	80					83			92	[135]
flake 3 × 5	marine water	H	205				5.33 × 10^7^/h	75			96.0	[85]
flake	ZnAc	G	265	6.67 × 10^3^ L/mol/min				92.0				[216]
pellet	Ionic liquid	G	190	0.16/min				51.6				[217]
PET	EG	G					99783/min	46.2				[207]
pellet 0.25	Na_2_CO_3_	G	196	3.767 L^2^/mol^2^/h		0.36 L/mol		185	80			[101]
fiber	Zn/Al	G	196					79.3	92			[190]
PET	KOH	M	65	0.0132	0.0246			127.6	96.0		96.0	[218]
PET	CH_3_OH	M	270	0.0033/s	0.0008/s							[206]
BHET	CH_3_OH	M	25	0.0017/day	0.0025/ day			56				[183]
PET	CH_3_OH	M	140	1.4 × 10^−3^ g PET/(mol min)			107.1 g PET/(mol min)	95.31				[219]
PET	CH_3_OH	M						66.5		80		[98]

H: hydrolysis; M: methanolysis; P: pyrolysis; Flake 3 × 5: flakes of 3 mm × 5 mm, Ea: activation energy; A: preexponential factor; 3Bu6DPB: tributylhexadecylphosphonium bromide.

**Table 5 polymers-13-01475-t005:** The various kind of reactor of recycling PET with product distribution.

Type of Reaction	Reactor	T (°C)	Catalyst	Yield (%)	Ref
Pyrolysis	Parr mini betch top reactor	500	NA	15	[221]
Liquefaction	Parr mini betch top reactor	500	NA	Single ring aromatic 11.1% 2nd ring polycyclic aromatic hydrocarbons 13.9% 3rd ring polycyclic aromatic hydrocarbons 7.7%	[221]
Pyrolysis	Boiling Flask	405	Ca(OH)_2_	ring aro Single matic 8.2% 2nd ring polycyclic aromatic hydrocarbons 11.4% 3rd ring polycyclic aromatic hydrocarbons 2.4%	[222]
Alcoholysis	Autoclave	205	Blank	DBTP (0%); EG (0%)	[223]
Alcoholysis	Autoclave	205	ZnCl_2_	DBTP (87.5%); EG (88.1%)	[223]
Alcoholysis	Autoclave	205	Zn(CH_3_COO)_2_	DBTP (88.4%); EG (88.6%)	[223]
Alcoholysis	Autoclave	205	Ti(CH_3_CH_2_CH_2_CH_2_O)_4_	DBTP (89.7%); EG (89.9%)	[223]
Alcoholysis	Autoclave	205	H_2_SO_4_	DBTP (67.3%); EG (67.2%)	[223]
Alcoholysis	Autoclave	205	(HO_3_S-(CH_2_)_3_-NEt_3_)Cl	DBTP (25.8%); EG (26.1%)	[223]
Alcoholysis	Autoclave	205	(HO_3_S-(CH_2_)_3_-NEt_3_)Cl-ZnCl_2_ (x = 0.67)	DBTP (95.3%); EG (95.7%)	[223]
Alcoholysis	Autoclave	205	(HO_3_S-(CH_2_)_3_-NEt_3_)Cl-FeCl_3_ (x = 0.67)	DBTP (85.8%); EG (85.7%)	[223]
Alcoholysis	Autoclave	205	(HO_3_S-(CH_2_)_3_-NEt_3_)Cl-FeCl_2_ (x = 0.67)	DBTP (85.5%); EG (85.6%)	[223]
Alcoholysis	Autoclave	205	(HO_3_S-(CH_2_)_3_-NEt_3_)Cl-CuCl_2_ (x = 0.67)	DBTP (77.8%); EG (77.9%)	[223]
Methanolysis	Autoclave	200	Alumunium triisopropoxide	DMT (88%)	[224]
Methanolysis	Autoclave	250–270	Zinc Acetate	DMT (60–95%)	[225]
Methanolysis	Microwave	160–200	Zinc Acetate	DMT	[226]
Hydrolysis	Vessel	115–145	((CH_3_)_3_N(C_16_H_33_))_3_(PW_12_O_40_)	TPA(90%)	[227]
Hydrolysis	Autoclave	70–95	TOMAB	TPA(98%)	[95]
Hydrolysis		190	Hydrotalcite	TPA(99%)	[228]
Hydrolysis	Microwave	90–98	TBAB	TPA(99%)	[133]
Alkaline Hydrolysis	Bottle neck	90	TBAI	TPA(90%)	[229]
Hydrolysis	Microwave	180	TOMAB	TPA(180%)	[230]
Aminolytic	NA	225–227 (m.p)	EA (pKa: 9.5)	BHETA (93%)	[114]
Aminolytic	NA	133–135 (m.p)	AEE (pKa:9.62)	BHEETA(92%)	[114]
Aminolytic	NA	152–154 (m.p)	AEAE (pka:9.62 and 6.49)	BHEAETA (55%)	[114]
Aminolytic	NA	220–222 (m.p)	AMP (pka:9.88)	BHMPTA (60%)	[114]
Aminolytic	NA	174–176 (m.p)	NMEA (pka:9.95)	BHEDMTA (87%)	[114]

## Data Availability

Not Applicable.

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
