# Peer review of "Strategic Possibility Routes of Recycled PET"

_polymers, 2021, doi:10.3390/polym13091475_

Round 1
Reviewer 1 Report
Presented work should be significantly changed and completed before the resubmission:
- The title should be changed. "Kinetic study" is not a main topic of the work. Moreover, chemical methods were described as well as mechanical methods. Authors should rebuild and complete their work.
- Recycling of poly(ethylene terephthalate) is well described in literature (research articles, reviews, books), so Authors should present novel point of view in this area. Review of proces kinetics and related issue is a good starting point. Current kinetics description is rather poor and not well organized (but the presentation of data in Tables and Figures is a good choice for the review)
- Main factors affecting the kinetics and obtained products should be described for each method. The type and amount of catalyst should be considered on the examplary works (for example: 10.1016/j.cej.2011.01.031, 10.1007/s12221-015-1213-4 and many others)
- Some informations about food grade recycled PET should be introduced (including the process description)
- Figure 4: 1.degradation (It should be indicated that m < n; please give examples of degradation factors, for example temperature or UV radiation) / 2. metholysia (It should be alcoholysis or methanolysis) / 7. Is it hydrolysis?
- Please, give examples of chemical recycling products applications (for example, some glycolysis products can be applied as polyols in polyurethanes synthesis)
- Pyrolysis (mechanism, products and kinetics) of PET should be described in details
- The detailed mechanism of each chemical method should be presented (including the effect of catalyst etc.).
Reviewer 2 Report
The review article discussing PET recycling, which is an important and timely topic. The manuscript is detailed but some important works and concepts are missing, and the organization of the manuscript needs to be improved. More details regarding major and minor corrections that are necessary to be addressed are listed below:
1) If the journal guideline permits, the authors should definitely add a table of contacts at the beginning of the manuscript. The manuscript is very long and has numerous sections and subsections that are impossible to follow, and a table of contents would be highly beneficial for the readers to navigate within the manuscript.
2) The title needs to be more specific. ‘new approach’ in the title does not have any information content. Be more specific.
3) The review title starts with ‘kinetic study’ but the shortest part of the manuscript is the kinetic recycling. Either elaborate on the kinetics more and deepen the discussions or change the emphasis in the title.
4) Aminolytic upcycling of poly(ethylene terephthalate) wastes using a thermally-stable organocatalyst should be mentioned in the review (10.1039/D0PY00067A).
5) Some of the critical aspects mentioned in the below article should be discussed in the review in order to increase the critical valuation in the review: https://cen.acs.org/environment/recycling/Plastic-problem-chemical-recycling-solution/97/i39
6) Briefly the upcycling of PET should be mentioned as a promising approach for PET recycling, few examples will suffice (10.1021/acsapm.9b00493; 10.1002/est2.201; 10.1101/2020.03.16.993592).
7) Reclaimed PET combination with bio-based monomers enabling plastics upcycling should be briefly discussed (10.1016/j.joule.2019.01.018).
8) Biological valorization of PET is a promising approach that is taking off at the moment and at least a few studies on this should be mentioned in the review (10.1021/acssuschemeng.9b03908).
9) Evaluating scenarios toward zero plastic pollution should be also briefly mentioned (10.1126/science.aba9475).
10) The figure captions are quite short. 1-2 lines figure captions with more details would guide the readers and aid understanding. Elaborate in the figure and table captions accordingly.
11) The conclusion section is very short and vague. The authors need to provide an appropriate conclusion based on the review and the hundreds of articles reviewed, as well as give their own opinion about the trends and future directions of the field.
